# ScaLA: Speeding-Up Fine-tuning of Pre-trained Transformer Networks via Efficient and Scalable Adversarial Perturbation

## Abstract

The size of transformer networks is growing at an unprecedented rate and has increased by three orders of magnitude in recent years, approaching the trillions. To train models of increasing sizes, researchers and practitioners have employed large-batch optimization to leverage massive distributed deep learning systems and resources. However, increasing the batch size changes the training dynamics, often leading to generalization gap and training instability issues that require extensive hyperparameter turning to maintain the same level of accuracy. In this paper, we explore the steepness of the loss landscape of large-batch optimization and find that it tends to be highly complex and irregular, posing challenges to generalization. To address this challenge, we propose ScaLA, a scalable and robust method for large-batch optimization of transformer networks via adversarial perturbation. Moreover, we perform several optimizations to reduce the computational cost of performing the adversarial perturbation, thereby improving its performance and scalability in the distributed training environment. We provide a theoretical convergence rate analysis for ScaLA using techniques for analyzing non-convex saddle-point problems. Finally, we perform an extensive evaluation of our method using BERT and RoBERTa on GLUE datasets. Our results show that our method attains up to $18 \times$ fine-tuning speedups on 2 DGX-2 nodes, while achieving comparable and sometimes higher accuracy than the state-of-the-art large-batch optimization methods. When using the same number of hardware resources, ScaLA is $2.7$–$9.8\times$ faster than the baselines.

## 1 Introduction

We have seen exponential growth in DL model size since the debut of the transformer network (Vaswani et al., 2017). For example, while BERT$_{base}$ has around 100M parameters, the model size has increased to multi-billion parameters such as Megatron-LM (8B) (Shoeybi et al., 2019), T5 (11B) (Raffel et al., 2019), Turing-NLG (17B) (tur), and with GPT-3 hitting a new staggering record of 175B parameters. With the three orders of magnitude growth, these large models also have powered accuracy breakthroughs in many challenging Natural Language Processing (NLP) tasks such as the General Language Understanding Evaluation (GLUE) benchmark (Wang et al., 2019a). Recent studies show that the performance of these models continues to scale with their sizes (Kaplan et al., 2020). As a result, we expect that the model size would continue to grow in the future.

To accelerate the training speed of large models, the most common way is to increase the batch size in the optimization algorithm in order to leverage multi-GPU training (Li et al., 2020; Liu et al., 2019; Huang et al., 2019; Shazeer et al., 2018; Shoeybi et al., 2019; Rajbhandari et al., 2019). By increasing the batch size, a mini-batch of size $B$ can be divided across more workers (GPUs), where the gradients are computed locally on each worker using back-propagation and then aggregated. Furthermore, most of the operations used in transformer networks are highly optimized in modern linear algebra frameworks on GPUs and can scale to larger batch sizes without significantly increasing the time per step (Wang et al., 2019c; Kaplan et al., 2020). If researchers can train each neural network with more GPUs and increased throughput, then it makes it possible for them to achieve better results by training even larger models, using larger datasets and exploring new ideas more rapidly.

However, changing the batch size is not always straightforward, as it often impacts the training dynamics. You et al. propose LAMB (You et al., 2019a) to exploit large-batch optimization for transformer networks. LAMB is a variant of Adam (Kingma & Ba, 2015) that applies layer-wise normalization before applying each gradient update, which has been used to successfully train BERT on 1024 TPU chips in 76 minutes. Despite showing promising results, prior work (You et al., 2019a) primarily focuses on pre-training. On the other hand, the fine-tuning stage starts to become a bottleneck (e.g., it takes tens of hours to fine-tune RoBERTa-large on MNLI (Wang et al., 2019a)) and becomes more expensive as model size increases. If we can speed-up pre-training by increasing batch sizes, *why do we not also increase the batch size during fine-tuning in the interest of making fine-tuning more efficient as well*? Contemporary experience is that fine-tuning with large batch sizes is harder to train, often reaching lower accuracy than the baseline accuracy using small batch sizes.

To address these challenges, we develop new approaches to improving the scalability and generalizability in fine-tuning pre-trained transformer networks by making the following contributions: (1) We present an adversarial perturbation based large batch optimization algorithm ScaLA (**Sca**lable **L**arge-batch **A**dversarial Perturbation) for training transformer networks, in the distributed training setting. We show how adversarial perturbation helps improve the generalization and more importantly how we reduce the cost of injecting adversarial perturbations to improve computational efficiency. (2) We also present a theoretical convergence rate analysis using techniques for analyzing non-convex saddle-point problems. (3) We conduct evaluation on a wide range of natural language understanding (NLU) tasks and assess the impact of adversarial perturbations on both the scalability and the generalizability in large batch task-specific fine-tuning. (4) We evaluate our approach against BERT (Devlin et al., 2019) and RoBERTa (Liu et al., 2019) and show that ScaLA obtains significant improvements over the state-of-the-art algorithms, such as LAMB, for the large batch optimization of fine-tuning tasks. Concretely, while LAMB leads to 1 point accuracy drop on average (e.g., GLUE) as we increase the batch size, our approach achieves the same and sometimes higher accuracy (up to 0.9 points) after drastically increasing the batch size. Furthermore, with our cost-efficient optimizations, ScaLA achieves up to $18\times$ speedups on 2 NVIDIA DGX-2 nodes over the baseline fine-tuning time and is up to $9.8\times$ faster than the baseline when using the same number of GPUs.

## 2 BACKGROUND AND RELATED WORK

Despite the great success of pre-trained transformer networks such as BERT (Devlin et al., 2019), a big challenge, in general, comes from the training efficiency – even with self-attention and parallelizable recurrence (Vaswani et al., 2017), and high-performance hardware (Jouppi et al., 2017), training transformer networks can still take a significant amount of time. One effective approach to reducing training time is through data parallelism (Devlin et al., 2019; Liu et al., 2019; Shoeybi et al., 2019), which motivates studies on large-batch stochastic non-convex optimizations for transformer networks (You et al., 2019a). These studies have raised concerns with respect to its convergence, generalizability, and training stability by observing that training with a large batch could be difficult (Keskar et al., 2017; Hoffer et al., 2017; Nado et al., 2021). Furthermore, prior works mostly focus on reducing the pre-training time (You et al., 2019a; Zhang & He, 2020; Gong et al., 2019; Clark et al., 2020) instead of the adaptation time at the fine-tuning stage.

While many researchers and practitioners focus on how to reduce the pre-training time, few attention has been paid to accelerate the fine-tuning stage, which gradually becomes a bottleneck as model sizes increase (e.g., it takes tens of hours to fine-tune MNLI on RoBERTa-large). Therefore, in contrast to previous works, our goal in this paper is to speed-up task-specific fine-tuning without hurting model accuracy. Different from pre-training, the fine-tuning stage often employs a much smaller batch size (e.g., $\log_2 B = \{4, 5\}$) than pre-training (e.g., $\log_2 B \geq 10$) (Devlin et al., 2019; Liu et al., 2019). The small batch size results in inefficient data parallelism (i.e., sub-optimal computation-communication ratio), making it difficult for fine-tuning to benefit from multi-GPU training. Moreover, a common understanding is that small-batch sizes provide implicit regularization effects (e.g., from gradient noise) that help improve generalization of downstream tasks. In contrast, our goal is to speed-up the fine-tuning process with large batch sizes while preserving model accuracy.

On a separate line of research, adversarial training was first proposed in the computer vision literature to improve a model's robustness against adversarial attacks (Goodfellow et al., 2015; Madry et al., 2018). Recently, there has been some work that shows that adversarial training helps improve model generalizability (Cheng et al., 2019; Wang et al., 2019b; Jiang et al., 2020; Liu et al., 2020; Yao et al.,

2018a; Zhu et al., 2020). However, very few works examine large-batch optimization of transformer networks and NLP tasks with adversarial perturbations from a computational efficiency and scalability perspective. The work most similar to ours is Zhu et al. (2020), who study adversarial training for NLP tasks by accumulating the gradient of the parameters from each of the ascent steps and updates the parameters only once after $K$ inner ascent steps with the accumulated gradients. Unlike Zhu et al. (2020), we consider accelerating the fine-tuning speed by parallel adversarial training and by adjusting the number of inner maximization steps, which offers much higher speedups.

## 3    THE PROPOSED METHOD

In this section, we present a principled method for large batch optimization that is highly scalable while maintaining the quality of the solutions as measured by task-appropriate accuracy metrics.

### 3.1    A SEQUENTIAL GAME-THEORETIC METHOD VIA ADVERSARIAL PERTURBATION ORACLE

**Formulation:** Let $\mathbb{X}$ denote the parameter space and $\mathbb{Y}$ denote the data (mini-batch/sample) space and $Q$ denote a distribution supported on $\mathbb{Y}$. To improve the generalizability of transformer fine-tuning while retaining the scalability, we augment the usual stochastic optimization objective by constructing an adversarial (Keskar et al., 2017; Madry et al., 2018) regularization. In particular, we solve the following robust optimization problem, which is a stochastic minimax (Lin et al., 2020) optimization problem augmented with a regularization term involving a deterministic adversarial perturbation, instead of vanilla risk minimization:

$$\min_{x \in \mathbb{X}} \mathbb{E}_{\xi \sim Q}[g(x, \xi)] = \min_{x \in \mathbb{X}} \mathbb{E}_{\xi \sim Q}[\underline{f}(x, \xi) + \lambda \underline{r}(x)]$$

$$= \min_{x \in \mathbb{X}} \max_{y \in \mathbb{Y}} \mathbb{E}_{\xi \sim Q}[\underline{f}(x, \xi) + \lambda r(x, y)] := \min_{x \in \mathbb{X}} \max_{y \in \mathbb{Y}} \mathbb{E}_{\xi \sim Q}[f(x, y, \xi)] \quad (1)$$

where $g : \mathbb{X} \times \mathbb{Y} \to \mathbb{R}$ denotes the robust training objective, $\underline{f} : \mathbb{X} \times \mathbb{Y} \to \mathbb{R}$ denotes the standard training objective, $f : \mathbb{X} \times \mathbb{Y} \times \mathbb{Y} \to \mathbb{R}$ denotes the augmented objective, $\underline{r} : \mathbb{X} \to \mathbb{R}$ denotes a deterministic regularization term on the parameters controlled by a strength factor $\lambda \in (0, \infty)$, $r : \mathbb{X} \to \mathbb{R}$ denotes the augmented regularization and $\xi$ denotes samples drawn from $Q$ (for simplicity, we slightly abuse the notation in using $\xi$ to denote the random variable, e.g. $\mathbb{E}_{\xi}[g(x, \xi)]$, or its empirical realizations, e.g. $\frac{1}{K} \sum_{k=1}^{K} g(x, \xi_k)$ for any $K$; the meaning is clear from the context). The overall (outer) training objective involves a minimization problem in the parameter space while being stochastic with respect to the data space. The adversarial regularization (inner) term is a deterministic maximization problem operating in the data space conditioned on a fixed parameter configuration. We wish to emphasize that this formulation is a two-player sequential (Jin et al., 2020), not simultaneous, game wherein the goal is to optimize a transformer network that is robust to adversarial perturbation. In a given round, the first player (associated with the outer minimization) proposes a parameter configuration, and the second player (associated with the inner maximization) responds with a penalty to capture the effect of label errors due to perturbations in a large data batch size to undermine the performance of the transformer parameter configuration chosen by the first player.

**Practical Considerations:** Language expressions are quite sensitive to individual words or clauses, where perturbations against those would likely generate incorrect or biased training data with wrong labels (Zhang & Yang, 2018). Following prior success in applying adversarial training to NLP models (Miyato et al., 2017; Zhu et al., 2020), we apply perturbations to the continuous word embeddings instead of directly to discrete words or tokens. The term $\underline{r}$ captures the prediction deviation from the perturbation. In a given round of the game, with respect to the first player's proposal, let $\Phi$ denote the transformer network under consideration (specifically, $\Phi$ is BERT in this paper) and $\xi$ be a large batch of data sampled from $Q$. We construct a label for the second player as $\gamma := \Phi(x, \xi)$. Next, for classification tasks, we choose $r$ to be the symmetric KL divergence (Jiang et al., 2020), i.e., $r(x, y) := \text{KL}_{\text{sym}}(\gamma, \Phi(x, y))$. We use symmetric KL divergence to measure the distributional divergence to generate adversarial perturbation. For regression tasks, we choose $r$ to be the squared loss, i.e., $r(x, y) := (\gamma - \Phi(x, y))^2$. In practice, we add an $\ell_\infty$ constraint on $y$, which is achieved by simple clipping with a radius of $\omega$ (projection). Intuitively, a large $r$ corresponds to a situation wherein the transformer is highly sensitive to a given perturbation in the input, suggesting that the model parameters are close to a sharp minimum. Augmenting the original training objective with $r$ makes the first player incur an additional penalty if the outer minimization solution veers closer to sharp minima, thereby encouraging flatter solutions and better generalizability.

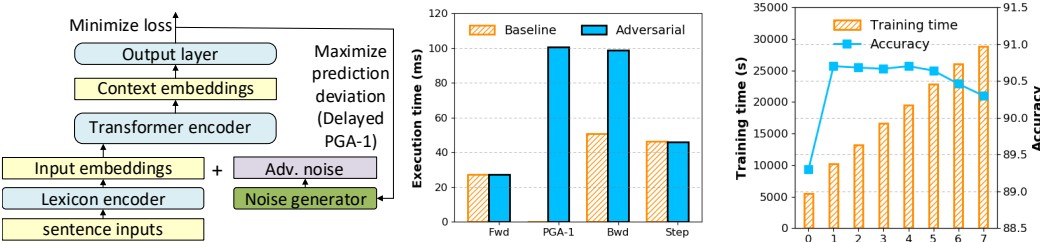

Figure 1: The architecture of the proposed method.

Figure 2: Time breakdown with and with PGA-1.

Figure 3: Impact of perturbation steps.

**Inner Maximization:** For any given outer step $t$, let $x_t$ denote the parameter proposed by the first player. Since the exact inner maximization in Equation equation 1 is intractable for non-convex models such as transformers, we adopt truncated methods as in prior works. Specifically, we use Projected Gradient Ascent (PGA) (Madry et al., 2018; Jiang et al., 2020) to solve this problem, i.e., $y_{\tau+1} = \Pi_\omega(y_\tau + \rho_\tau \nabla_y r(x_t, y))$ where $\rho_\tau$ for $\tau \in [\mathcal{T}]$ is the step size sequence and $\Pi$ projects the result of the gradient ascent update into an $\ell_\infty$ ball of diameter $2\omega$ around the original input embeddings, $\xi$, considered by the first player.

**Outer Minimization via Groupwise Adaptive Learning Rates:** Inspired by prior works that stabilize large-batch training and improve its convergence quality, we employ algorithms with groupwise adaptive learning rates (You et al., 2019a) to solve the outer minimization. Specifically, we solve the minimization problem in Equation equation 1 by $x_{t+1}^i = x_t^i - \eta_t \nu(\|x_t^i\|)\widehat{\nabla}_x^i g(x)/\|\widehat{\nabla}_x^i g(x)\|$, $\forall i \in [h]$ where $i$ denotes the $i^{\text{th}}$-layer of the transformer. The normalized gradient descent mitigates issues due to exploding gradients. The learning rate sequence $\eta_t, \forall t \in [T]$ is scaled by a clipping function $\nu(c) := \max(\mathcal{L}, \min(c, \mathcal{U}))$ where $\mathcal{L} < \mathcal{U}$ (e.g., $\mathcal{L} = 0$ and $\mathcal{U} = 10$), which ensures the norm of the update is of the same order as that of the weights. Note that we use gradient averaging on $\xi$, i.e., gradient accumulation and all-reduce, over a batch size $B$ distributed across $P$ workers in order to obtain a noisy gradient estimate $\widehat{\nabla}_x^i g(x)$ at epoch $t$.

**Computational Cost:** Given that the primary interest of using a large batch size is to improve hardware efficiency, we are motivated to look carefully into the computational cost of adversarial perturbation. Adversarial perturbation requires an extra PGA inner loop that standard training does not have. Figure 2 provides the time breakdown of adversarial training using PGA with $\mathcal{T} = 1$ (denoted as PGA-1). PGA-1 performs the perturbation and takes approximately the same time as making three forward passes (Fwd) through the network. This is because one step of PGA requires to make one forward and backward pass (Bwd) over the entire network. The backward pass of adversarial training takes roughly twice the amount of time as the standard backward step because the back-propagation is triggered twice to calculate the perturbation noise and the gradients. The time spent on the optimizer step function (step) remains the same. In total, adversarial training slows down training by at least 2 times, even with $\mathcal{T}=1$. This motivates us to look at the effectiveness of different perturbation steps as well as the usefulness of perturbation from the initial epochs in practice.

**Impact of Perturbation Steps, $\mathcal{T}$:** Prior works often do multiple gradient computation steps ($\mathcal{T} > 1$) and take several times longer training time to produce adversaries (Madry et al., 2018; Zhu et al., 2020), because their focus is not on computational efficiency. Subsequently, researchers presented Curriculum Adversarial Training (CAT) (Cai et al., 2018) and Annealing-based Adversarial Training (Amata) (Ye et al., 2020), which progressively increase the perturbation with various strengths, cutting the adversarial training cost while maintaining good accuracy. To investigate how CAT and similar methods affect large-scale NLP problems involving transformers, we evaluate the final fine-tuning accuracy and training cost of QNLI, varying the number of perturbation steps $\mathcal{T}$ and report the results in Figure 3. Interestingly, although using a large $\mathcal{T}$ helps to produce stronger adversaries, we find that this does not lead to improved fine-tuning accuracy, despite the fact that the training overhead still increases almost linearly. In fact, the best accuracy is achieved with $\mathcal{T} = 1$.

Our hypothesis to this phenomenon is below. The model has two components, namely, the parameter space and data space. First, unlike the minimization in the parameter space, which is stochastic, the maximization in the data space is deterministic. Second, with respect to the testing phase,

the numerical convergence in the model's parameter space is of primary importance rather than the numerical convergence in the data space, i.e., the maximization is an auxiliary procedure that augments the training phase to make the parameter space "aware" of effects of the batch size across epochs. Due to these two points, at a certain epoch, for a given batch, the marginal utility of an additional PGA step is low, and we are able to get away with inexact deterministic maximization. Therefore, we apply PGA-1 in our large-batch optimization scheme, given that it produces sufficiently good solutions while being much more computationally efficient.

**Delayed Perturbation Injection:** Given that even PGA-1 still adds an overhead factor of 2, we are motivated to investigate how useful adversarial perturbations are in the initial phase of large-batch optimization. We conduct additional experiments to measure the final accuracy corresponding to starting from a regular fine-tuning and then enabling PGA-1 for $t \geq t_s$ where $t_s \in [T]$. Our observation is that enabling PGA-1 from the beginning does not offer much improvement in accuracy, whereas adversarial perturbation becomes more potent as the model begins to stabilize towards the end of training. Intuitively, this makes sense because generally, at initialization, the model's parameters are relatively far from their final values and are less likely to get stuck at local minima. Therefore the adversarial perturbations generated in the initial training iterations are quite different from the perturbations towards the end of training because they would not maximize the adversarial loss in Equation 1. We remark that a similar phenomenon has been observed in computer vision tasks (Cai et al., 2018; Ye et al., 2020; Gupta et al., 2020). We show that it is possible to delay the injection of adversarial perturbations for large-batch optimization of transformers for NLP tasks.

**The Algorithm, ScaLA:** Combining the formulation with the above investigations, we construct our distributed large-batch transformer fine-tuning algorithm, named ScaLA (Algorithm 1), whose convergence rate is characterized in Theorem 1.

---

**Algorithm 1** , ScaLA, **Sca**lable **L**arge-batch **A**dversarial Perturbation

---

1: **Input:** Epochs $T$, delay $t_s$, perturbation (inner) step size $\rho$, clipping radius $\omega$, regularization strength $\lambda$, (outer) learning rate $\eta$
2: **Output**: $h$-layer transformer model $\Phi$ with converged robust parameters $\overline{x} := x_T$
3: **for** $t \in [T]$ **do**                                                  ▷ Loop through epochs
4:     **for** worker $p \in [P]$ **do**                          ▷ In parallel across homogeneous workers
5:         **for** mini-batch $\xi_p \sim Q$ **do**             ▷ Subsample $\frac{B}{P}$ data instances on each worker
6:             $\underline{r}(x_t) \leftarrow 0, \gamma \leftarrow \Phi(x, \xi_p)$, select $y_0$             ▷ Initialize regularization and label
7:             **if** $t \geq t_s$ **then**                                    ▷ Check delay condition
8:                 $y_1 \leftarrow \Pi_\omega(y_0 + \rho \nabla_y r(x_t, y))$     ▷ Perform adversarial perturbation with PGA-1
9:                 $\underline{r}(x_t) \leftarrow \text{KL}_{\text{sym}}(\gamma, \Phi(x_{t-1}, y_1))$     ▷ Calculate the adversarial regularization
10:            $g(x_t, \xi_p) \leftarrow \underline{f}(x_{t-1}, \xi_p) + \lambda \underline{r}(x_t)$             ▷ Calculate the augmented loss
11:            $\nabla_x g(x_t, \xi_p) \leftarrow$ Backward pass on $\Phi$ ▷ Compute local gradients using accumulation
12:     $\widehat{\nabla}_x g(x_t) \leftarrow \frac{1}{B} \sum_{p=1}^{P} \nabla_x g(x_t, \xi_p)$                     ▷ Gradient averaging using all-reduce
13:     $x_t^i \leftarrow x_{t-1}^i - \eta_t \nu(\|x_t^i\|) \frac{\widehat{\nabla}_x^i g(x_t)}{\|\widehat{\nabla}_x^i g(x_t)\|}, \forall i \in [h]$                     ▷ Update model parameters

---

**Theorem 1** (Complexity of Algorithm 1; Informal – Details in Appendix D). *Consider the problem in Equation 1. Let $t_s = 0$. Setting the outer learning rate as $\eta = O\left(1/\sqrt{T}\right)$ and scaling batch size as $b = O(T)$, for Algorithm 1, we have $\mathbb{E}\left[\|\nabla g_{1/2\alpha}(\overline{x})\|^2\right] \leq O\left(\epsilon + \kappa_\alpha/\sqrt{T}\right)$ where $\overline{x}$ is the estimator obtained from running $T$ steps of Algorithm 1 and picking $x_t$ uniformly at random for $t \in [T]$. Here, $\epsilon$ is the error due to the approximate inner maximization oracle, $\alpha$ characterizes the smoothness of $f(x,.)$, $g_{1/2\alpha}$ is the Moreau-envelope of $g$ and $\kappa_\alpha = \max_i \alpha_i / \min_i \alpha_i$.*

## 4  EVALUATION

We evaluate the effectiveness of ScaLA in training transformer networks over a set of NLP tasks.

**Hardware:** We study the efficiency of computation using 2 NVIDIA DGX-2 nodes. Each node consists of 16 NVIDIA V100 GPUs. The nodes are connected with InfiniBand using a 648-port Mellanox MLNX-OS CS7500 switch. We vary the number of workers (i.e., from 1 GPU to 32 GPUs) in the experiments to evaluate the scalability. **Software:** We use PyTorch Distributed Data Parallel to

scale the fine-tuning from a single GPU to multiple GPUs, which uses all-reduce (Proficz, 2018) to compute the average of the gradients across all workers for a parameter update. We use NCCL V2.4 as the underlying all-reduce implementation. **Model/Dataset:** We use pre-trained BERT$_{base}$ model released by HuggingFace (Wolf et al., 2020). We use the GLUE benchmark (Wang et al., 2019a), which is a collection of sentence or sentence-pair natural language understanding tasks including question answering, sentiment analysis, and textual entailment. Given that large models normally require days or even weeks (e.g., RoBERTa) if trained from scratch, it is not cost-efficient to evaluate from scratch using large-scale pretraining. Therefore, experiments presented in this section focus on fine-tuning pre-trained models, e.g., BERT (Devlin et al., 2019) and RoBERTa (Liu et al., 2019), and exclude fine-tuning tasks that have very small datasets (e.g.,CoLA, RTE). We report the details about the hyperparameters in Appendix B.

### 4.1 MAIN RESULTS – TRAINING TIME ACCELERATION AND ACCURACY IMPROVEMENT

We first compare the following schemes: (1) **Single GPU + SB:** This is the existing PyTorch implementation of Transformer fine-tuning from HuggingFace (HF), using small batch (SB) sizes (e.g., 32). Our fine-tuning results on single GPU achieves higher accuracy than what was reported by (Devlin et al., 2019), (2) **Multi-GPU + SB:** This is multi-node multi-GPU PyTorch fine-tuning implementation using DistributedDataParallel (Li et al., 2020), and (3) **Multi-GPU + LB + ScaLA:** This is our approach as described in Algorithm 1, using large minibatches (LB), e.g., 1K. Table 2 shows results on MNLI, QNLI, QQP, and SST2, which are larger datasets and less sensitive to random seeds. $n \times g$ refers to $P_n$ nodes each with $P_g$ GPUs for a total of $P = P_n P_g$ homogeneous workers (e.g., 32 GPUs on 2 NVIDIA DGX-2 nodes). For a fair comparison, we reproduce BERT and RoBERTa baseline. Our reproduced baseline achieves the same or slightly higher accuracy than the originally reported results in (Devlin et al., 2019) and (Liu et al., 2019). We now discuss our results and observations.

Table 1: The training time and accuracy results on GLUE benchmark. Results show that ScaLA is able to achieve the same average accuracy as the baseline while providing up to $18\times$ speedups than single GPU, and up to $9.8\times$ speedups when using the same amount of hardware resources.

| BERT$_{base}$ | n×g | bsz | MNLI-m | | | QNLI | | | QQP | | | SST-2 | | | Avg. |
|---|---|---|---|---|---|---|---|---|---|---|---|---|---|---|---|
| | | | Steps | Time | Acc. | Steps | Time | Acc. | Steps | Time | Acc/F1 | Steps | Time | Acc. | |
| Devlin et al. 2019 | | | | | 84.4 | | | 88.4 | | | - | | | 92.7 | - |
| Baseline (B=32) | 1x1 | 32 | 73632 | 19635 | 84.8 | 19644 | 5535 | **90.6** | 68226 | 16494 | **91/88.0** | 12630 | 2736 | 93.1 | **89.4** |
| Baseline (B=32) | 2x16 | 32 | 73632 | 8848 | 84.8 | 19644 | 2408 | 90.6 | 68226 | 11311 | 91/88.0 | 12630 | 1494 | 93.1 | **89.4** |
| ScaLA (B=1K) | 2x16 | 1K | 2301 | **1323** | **85.1** | 615 | **432** | 90.0 | 2133 | **4229** | 90.9/87.7 | 396 | **151** | **93.5** | 89.4 |

| RoBERTa$_{large}$ | n×g | bsz | MNLI-m | | | QNLI | | | QQP | | | SST-2 | | | Avg. |
|---|---|---|---|---|---|---|---|---|---|---|---|---|---|---|---|
| | | | Steps | Time | Acc. | Steps | Time | Acc. | Steps | Time | Acc/F1 | Steps | Time | Acc. | |
| Liu et al. 2020 | | | | | 90.2 | | | 94.7 | | | 92.2/- | | | 96.4 | - |
| Baseline (B=32) | 1x1 | 32 | 73632 | 43090 | 90.5 | 19644 | 14188 | 94.7 | 68226 | 40945 | 92.0/89.4 | 12630 | 4940 | 96.4 | 92.5 |
| Baseline (B=32) | 2x16 | 32 | 73632 | 18114 | 90.5 | 19644 | 4842 | 94.7 | 68226 | 16614 | 92.0/89.4 | 12630 | 3072 | 96.4 | 92.5 |
| ScaLA (B=1K) | 2x16 | 1K | 2301 | **3363** | **90.9** | 615 | **1168** | **95.1** | 2133 | **2404** | **92.3/89.8** | 396 | **401** | **96.7** | **92.9** |

**Improving scalability with ScaLA:** Compared with single-GPU training, baseline fine-tuning on multi-GPU leads to only modest training speedup improvements, e.g., with $1.5 - 2.4\times$ faster training speed for both BERT and RoBERTa, even with $32\times$ more compute resources. The speedup is limited because of the small mini-batches (e.g., 32) used for fine-tuning, which do not provide sufficient workload to fully utilize the underlying hardware (e.g., many cores stay idle). Thus, communication overhead becomes the dominant part, and fine-tuning tasks struggle to obtain speedups with more than 4 GPUs (see detailed analysis results later). In contrast, ScaLA achieves up to $18\times$ speedups with 32 GPUs. When using the same number of GPUs (e.g., 32), ScaLA is 2.7–9.8$\times$ faster than Baseline ($B = 32$). The speedup is significant because (1) large batches prevent hardware from under-utilization and enables processing more samples per second; (2) it takes fewer iterations to process an epoch, hence the reduced all-reduce operations to exchange gradients and the overall increased computation-vs-communication ratio; (3) PGA-1 significantly reduces the cost to generate adversarial perturbations; and (4) With delayed perturbation injection, PGA-1 is only added to later iterations instead of throughout the entire training process, further reducing the training cost. Finally, ScaLA obtain the speedups while achieving the same accuracy (88.4 vs. 88.4) average accuracy for BERT and higher accuracy (92.9 vs. 92.5) for RoBERTa as the baselines.

**Closing generalization gap with ScaLA:** We also compare alternative methods that perform large-batch optimizations: (1) Multi-GPU + LB: This configuration uses large mini-batches (e.g., 1K), and applies heuristic-based scheduling rule (e.g., square root) and grid search for learning rates; (2) Multi-GPU + LB + LAMB: Applies the large-batch optimizer LAMB (You et al., 2019a) to fine-tuning tasks. First, compared with the baseline, the accuracy of Multi-GPU + LB drops by close to 1 point (88.4 vs. 89.4, and 92.1 vs. 92.9) in average and close to 2 points for some tasks (e.g., QQP on BERT), indicating that it is challenging to obtain on-par accuracy with large-batch optimizations for fine-tuning tasks. Second, LAMB leads to only marginal improvements (88.6 vs. 88.4, and 92.1 vs. 92.1) than the baseline and is 0.8 points lower than the small-batch baseline. Since LAMB performs no explicit regularization to model complexity, it may still lead to overfitting on downstream tasks. With ScaLA, we are able to close the generalization gap from large-batch optimization (89.4 vs. 89.4, and 92.5 vs. 92.9) and achieve 0.8 points higher accuracy (89.4 vs. 88.6, 92.9 vs. 92.1) than LAMB on both BERT and RoBERTa. ScaLA improves generalizability because it introduces adversarial perturbations, which serves as a regularizer. By training the network to be robust to such perturbations, the model loss landscape is smoothed out, leading to improved generalization.

Table 2: The comparison results of the GLUE benchmark. ScaLA outperforms large-batch optimization baselines by achieving higher accuracy after training the same number of samples.

| $BERT_{base}$ | n×g | Batch size | MNLI-m | | | QNLI | | | QQP | | | SST-2 | | | Avg. |
|---|---|---|---|---|---|---|---|---|---|---|---|---|---|---|---|
| | | | Steps | Time | Acc. | Steps | Time | Acc. | Steps | Time | Acc/F1 | Steps | Time | Acc. | |
| Baseline (B=1K) | 2x16 | 1K | 2301 | 1148 | 84.3 | 615 | 349 | 89.3 | 2133 | 2892 | 89.6/86.1 | 396 | 134 | 93 | 88.4 |
| LAMB (B=1K) | 2x16 | 1K | 2301 | 1180 | 84.1 | 615 | 359 | 89.6 | 2133 | 2978 | 90.5/87.0 | 396 | 139 | 92.4 | 88.6 |
| ScaLA (B=1K) | 2x16 | 1K | 2301 | 1323 | **85.1** | 615 | 432 | **90.0** | 2133 | 4229 | **90.9/87.7** | 396 | 151 | **93.5** | **89.4** |

| $RoBERTa_{large}$ | n×g | Batch size | MNLI-m | | | QNLI | | | QQP | | | SST-2 | | | Avg. |
|---|---|---|---|---|---|---|---|---|---|---|---|---|---|---|---|
| | | | Steps | Time | Acc. | Steps | Time | Acc. | Steps | Time | Acc/F1 | Steps | Time | Acc. | |
| Baseline (B=1K) | 2x16 | 1K | 2301 | 2514 | 90.1 | 615 | 936 | 94.3 | 2133 | 1874 | 91.7/89.1 | 396 | 317 | 95.9 | 92.1 |
| LAMB (B=1K) | 2x16 | 1K | 2301 | 2646 | 90.5 | 615 | 973 | 94.5 | 2133 | 1998 | 91.3/88.5 | 396 | 324 | 96.2 | 92.1 |
| ScaLA (B=1K) | 2x16 | 1K | 2301 | 3363 | **90.9** | 615 | 1168 | **95.1** | 2133 | 2404 | **92.3/89.8** | 396 | 401 | **96.7** | **92.9** |

## 4.2 EXPERIMENT – ANALYSIS RESULTS

**Ablation analysis:** In this part, we study the importance of components in ScaLA. We set $t_s$ to 0, which denotes as *w/o Delaying PGA-1*. We replace the outer minimization to use ADAM (Kingma & Ba, 2015), which is noted as *w/o Groupwise LR*. We set $\lambda$ to 0, which denotes as *w/o PGA-1*. The results are reported in Table 3.

Table 3: Ablation study of ScaLA using $BERT_{base}$ on GLUE tasks.

| | MNLI-m | | QNLI | | QQP | | SST-2 | | Avg. |
|---|---|---|---|---|---|---|---|---|---|
| | Time | Acc. | Time | Acc. | Time | Acc/F1 | Time | Acc. | |
| BERT | 19635 | 84.8 | 5535 | 90.6 | 16494 | 91/88.0 | 2736 | 93.1 | 89.4 |
| ScaLA | 1323 | 85.1 | 432 | 90 | 4229 | 90.9/87.7 | 151 | 93.5 | 89.4 |
| w/o Delaying PGA-1 | 2503 | 85.2 | 726 | 90.2 | 6407 | 91.3/88.3 | 272 | 93.1 | 89.5 |
| w/o Groupwise LR | 1290 | 85.0 | 422 | 89.9 | 4212 | 90.7/87.6 | 146 | 93.0 | 89.2 |
| w/o PGA-1 | 1180 | 84.1 | 359 | 89.6 | 2978 | 90.5/87.0 | 139 | 92.4 | 88.6 |

The results show that the removal of either design element would result in a performance drop. For example, removing groupwise learning rates leads to 0.2 points accuracy drop (89.2 vs. 89.4), while completely removing PGA-1 leads to 0.8 points accuracy drop (88.6 vs. 89.4). This result demonstrates that these two components are complementary to each other. If we perform PGA-1 without delayed injection, the average accuracy increases by 0.1 point (89.5 vs. 89.4), but the execution time is increased significantly. This difference will be particularly felt during pre-training or training larger models. With our delayed PGA-1 approach, we save the training time by 1.5–1.9× while retaining high accuracy.

**Scalability analysis varying GPUs.** We carry out a scalability test by varying the number of GPUs from 1 to 32, with and without communication. We choose a batch size 32, and divide the samples among $P$ GPUs. If the per-GPU batch size (e.g., 16) is larger than the maximum admissible per-GPU batch size (e.g., 8), we use local gradient accumulation (Goyal et al., 2017) to avoid running out of memory. Figure 4a shows the scalability results. For batch size 32, the training time decreases when the number of workers increases. However, it quickly plateaus and increases slightly after 4 GPUs. This is because small-batch leads to more frequent communication among workers. As a result, the

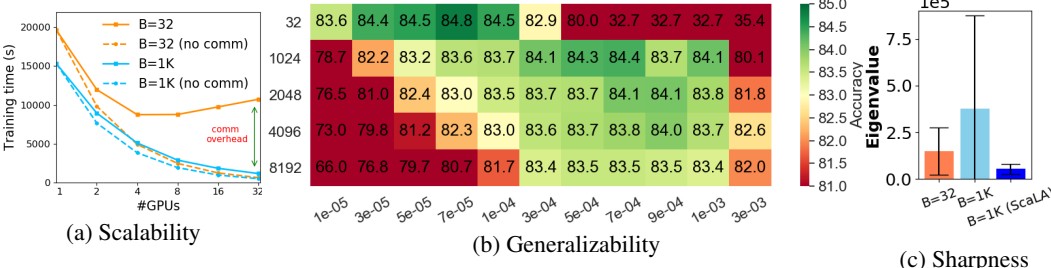

(a) Scalability        (b) Generalizability        (c) Sharpness

Figure 4: Scalability and generalizability results by fine-tuning BERT$_{base}$ using ADAM with different batch sizes on the MNLI task from the GLUE benchmark.

communication overhead dominates the total execution time and hinders the scalability of multi-GPU training. In contrast, by increasing the batch size, the training time keeps decreasing as the number of GPUs increases because a large batch not only reduces the number of expensive all-reduce operations for exchanging gradients (i.e., increased computation-vs-communication ratio) but also increases the training throughput per GPU due to increased computation granularity. Figure 5 shows the scalability comparison on SST-2 after optimizations. While the speedup still plateaus at 4 GPUs with a small batch size (e.g., $B = 32$), the four large-batch configurations are able to scale well up to 32 GPUs and take a similar amount of time with 32 GPUs. ScaLA scales better than ScaLA without delaying PGA-1, and achieves a much faster training speed, especially in the 1-16 GPU range.

**Batch sizes vs. learning rates for fine-tuning tasks.** Figure 4b reports the learning rate scaling effects on fine-tuning Transformer networks. We observe that the learning rate scales roughly with the the square root of the increase of the mini-batch size (That said, the Appendix C provides analysis to show that the best learning rates do not always follow sqrt rule). This is consistent with prior findings in pre-training (You et al., 2019a) and simplifies the hyperparameter tuning effort for ScaLA.

**Curvature analysis.** Prior work (Keskar et al., 2017) correlates the low generalization with *sharp* minima (which are characterized by a positive curvature of large magnitude in the parameter space). To verify this hypothesis, we quantitatively measure the steepness of loss landscape by loading the checkpoint of a converged model and computing the curvature, i.e., properties of the second derivative of the model, with respect to its parameters, for a fixed batch of samples. In particular, following (Yao et al., 2018b), for a model $\Phi(x)$, we compute the largest eigenvalue of the model's Hessian, $L_{\max}[\nabla_x^2 \Phi(x)]$, using the Hessian-vector product primitive and the power method. We use the largest eigenvalue as a measure of sharpness since the corresponding (top) eigenvector characterizes the direction of the largest change in gradient at a given point in the parameter space. From Figure 4c, the largest eigenvalue of the model trained with large batch (e.g., 1K) is much larger (e.g., 2.6x) with higher deviations (e.g., 3.9x) compared to the small batch baseline. Thus, we empirically find that large-batch optimization makes the loss landscape of the model more prone to ill-conditioning and less robust to perturbation, which may explain the loss in generalization. We then measure the steepness of the loss landscape again after applying ScaLA. As shown in Fig. 4c, the largest eigenvalue of the model becomes much smaller ($6.9\times$) with lower deviations with ScaLA and is slightly better than the small batch baseline, which is a strong indication that our approach enforces the smoothness of the model that leads to the accuracy improvement.

**Comparison with random noise.** We have performed additional experiments by creating perturbations through adding Gaussian noise to the embeddings. Table 5 that random noise indeed can improve the accuracy for MNLI-m (84.3 vs. 84.5), QNLI (89.3 vs. 89.4), and QQP (90.3/87.0 vs. 89.6/86.1) over the baseline, but it also leads to worse results on SST-2 (93. vs. 92.6). Compared with ScaLA, random noise consistently falls behind ScaLA in its ability to reduce the generalization error on all tested tasks and is on average 0.7 points lower than ScaLA (88.7 vs. 89.4).

**Perturbations via ground-truth vs. label probability.** We also create one-hot labels and use those to generate perturbations instead of using label probability generated by the network. Table 5 shows that using label probability (LP) consistently leads to higher accuracy than using the ground-truth (GT), e.g., 89.4 vs. 89.0 on average. Label probability leads to better generalization, probably because it provides a better measurement of the adversarial direction, which is the direction in the input space in which the label probability of the model is most sensitive to small perturbations.

Table 4: Evaluation results on alternative methods to generate perturbations using random noise, ground-truth, and label probability.

| Model | MNLI-m | QNLI | QQP | SST-2 | Avg |
|---|---|---|---|---|---|
| Baseline | 84.3 | 89.3 | 89.6/86.1 | 93 | 88.4 |
| Gaussian noise | 84.5 | 89.4 | 90.3/87.0 | 92.6 | 88.7 |
| ScaLA (GT) | 84.1 | 89.6 | 90.7/87.6 | 93.2 | 89.0 |
| ScaLA (LP) | **85.1** | **90** | **90.9/87.7** | **93.5** | **89.4** |

Table 5: Comparison results with FreeLb.

| | MNLI-m | | SST-2 | |
|---|---|---|---|---|
| | Acc. | Time | Acc. | Time |
| Baseline | 84.8 | 8848 | 93.1 | 2736 |
| FreeLb | **85.1** | 3773 | 93.3 | 389 |
| ScaLA | **85.1** | 1323 | **93.5** | 151 |

**Comparison with FreeLb.** We also compare ScaLA with FreeLb. The original FreeLb does not support multi-node training. We extend it with PyTorch DDP to train in a multi-node distributed training environment. We observe that although both FreeLb and ScaLA achieve similar accuracy, ScaLA is much faster than FreeLb. ScaLA is faster because FreeLb still performs multiple ascent steps to calculate adversaries cross the entire training. In contrast, ScaLA takes several optimizations to reduce the adversarial perturbation cost as well as leveraging group-wise adaptive learning rates to enable training with larger batch sizes, which improves the computational efficiency.

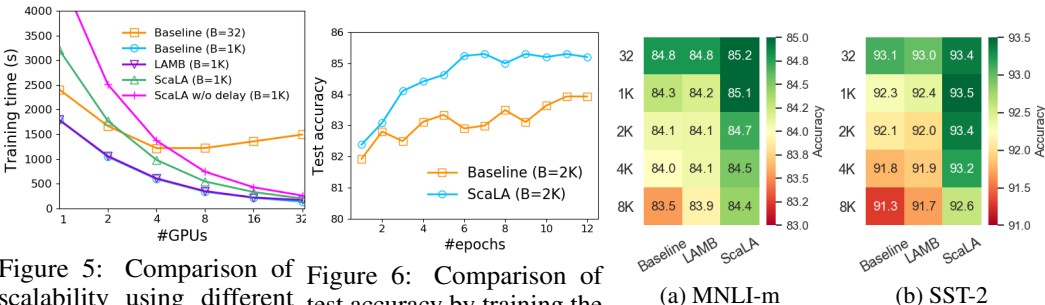

Figure 5: Comparison of scalability using different large-batch optimization methods on SST-2.

Figure 6: Comparison of test accuracy by training the baseline longer.

(a) MNLI-m  (b) SST-2

Figure 7: Comparison of accuracy under even larger batch sizes.

**Train longer, generalize better?** Despite improved training speed and accuracy, one may still wonder whether the generalization gap can be mitigated by training the downstream tasks longer. (Figure 6 shows the comparison results between ScaLA and the baseline on a batch size of 2K. ScaLA obtains an accuracy of 85.2 after 6 epochs of training, whereas the baseline has difficulty to reach 84 after training twice longer (e.g., 12 epochs). ScaLA achieves better accuracy because it explicitly penalizes model weights from getting stuck at sharp minima, leading to better generalizability.

**Generalizability under different batch sizes.** In this part, we evaluate how different batch sizes affect the generalizability of fine-tuning transformer tasks. Figure 7 shows the results on MNLI-m and SST-2. We make two major observations: (1) The accuracy tends to drop as the batch size increases. (2) While both the baseline and LAMB suffer from significant accuracy drop by drastically increasing the batch size (e.g., from 32 to 8K), ScaLA is able to mitigate the generalization gap and consistently achieves higher accuracy than the baseline (e.g., 84.4 vs. 83.5 for MNLI, and 92.6 vs. 91.3 for SST-2 at batch size 8K) and LAMB (e.g., 84.4 vs. 83.9 for MNLI, and 92.6 vs. 91.7 for SST-2 at batch size 8K). These results indicate the benefit of ScaLA is maintained by further increasing the batch size, which could bring even greater speedups when increasing the data parallelism degree.

## 5 CONCLUSIONS AND FUTURE DIRECTIONS

In this paper, we study how to add adversarial perturbations to improve the scalability and generalizability of training large transformer networks in a principled manner. We introduce ScaLA, a scalable and generalizable large-batch optimization method using adversarial perturbations. The experiment results show that ScaLA obtains up to $18\times$ speedups on fine-tuning transformer networks and outperforms the state-of-the-art large-batch optimization methods in accuracy. Despite offering great speedups without losing accuracy, ScaLA is limited in that it currently has only been evaluated against fine-tuning tasks of pre-trained transformer networks due to the expensive pre-training cost. Furthermore, it has not been tested on emerging pre-trained transformer networks for computer vision tasks, such as ViT, so it is unclear about the effectiveness of ScaLA on those tasks. We plan to explore how to generalize our proposed method to improve the generalization of the pre-training of large-scale language models as well as tasks in other domains in future work.

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

# A    ADDITIONAL RESULTS

In the part, we present results that are not included in the main text due to the space limit.

## A.1    THE USEFULNESS OF PERTURBATION IN THE INITIAL EPOCHS

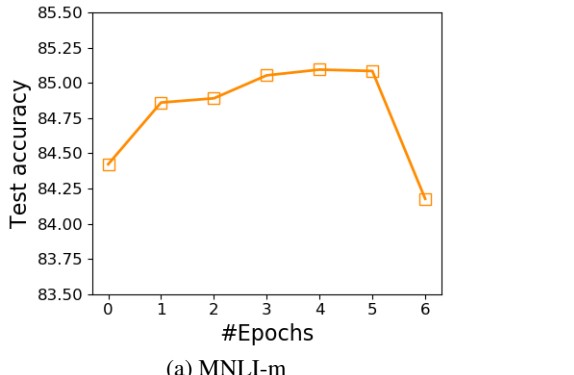
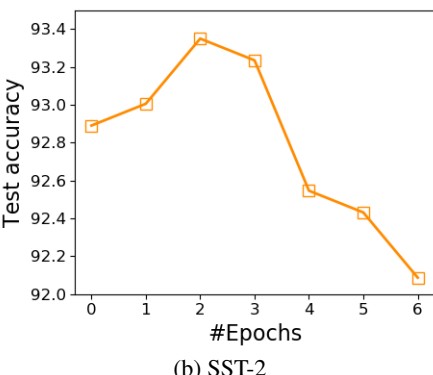

(a) MNLI-m                                   (b) SST-2

Figure 8: Accuracy results from delaying the injection of PGA-1 at different epochs.

In Section 3.1, we mention that no adversaries are needed at the initial epochs of fine-tuning. To verify, we conduct experiments to measure the final accuracy corresponding to starting from regular training and switching to PGA-1 after $t_s$ epochs, where $t_s \in [T]$. Figure 8 shows that enabling PGA-1 from the very beginning does not offer much improvement on accuracy. However, as we delay the adversarial perturbation, the model accuracy starts to increase. This is because, at initialization, the model's parameters are relatively far from their final values and are less likely to get stuck at local minima. By delaying the injection of adversarial perturbations, we observe improved test accuracy on fine-tuning tasks. However, it also seems that such perturbation should not be injected too late, which may inadvertently affect the accuracy. It is possible that a more advanced method to adaptively choose the value of $t_s$ is desired. However, given that (1) the primary focus of this work is to demonstrate that it is possible and effective to leverage adversarial training for large-batch optimization of transformer networks and (2) the search space of is quite small for fine-tuning tasks, we leave this as an interesting research question for future exploration.

# B    HYPERPARAMETERS

For all configurations, we fine-tune against the GLUE datasets and set the maximum number of epochs to 6. We use a linear learning rate decay schedule with a warm-up ratio of 0.1. For ScaLA, we set $\lambda = 1$, perturbation clipping radius $\omega = 10^{-5}$, step size $\rho = 10^{-4}$, and $t_s$={3,5}. These values worked well enough that we did not feel the need to explore more. For fairness, we perform a grid search of learning rates in the range of {1e-5, 3e-5, 5e-5, 7e-5, 9e-5, 1e-4, 3e-4} for small batch sizes and {5.6e-5, 8e-5, 1e-4, 1.7e-4, 2.4e-4, 2.8e-4, 4e-4, 5.6e-4, 1e-3} for large batch sizes. We keep the remaining hyperparameteres unchanged.

# C    LEARNING RATE SCALING FOR FINE-TUNING TASKS

In this part, we investigate how large-batch optimization affects the generalizability in fine-tuning transformer networks. As there are various heuristics for setting the learning rates (Smith et al., 2018; Goyal et al., 2017; Smith, 2018; You et al., 2019a), and because few work studies the learning rate scaling effects on fine-tuning Transformer networks, we perform a grid search on learning rates {1e-4, 3e-4,5e-4, 7e-4, 9e-4, 1e-3, 3e-3} and batch sizes {1K, 2K, 4K, 8K} while keeping the other hyperparameters the same to investigate how ScaLA affects the hyperparameter tuning effort for large batch optimizations.

Table 6 below shows the results of using the square root scaling rule to decide the learning rates for large batch sizes vs. accuracy results with tuned learning rate results, without and with ScaLA. The

first row represents the best accuracy found through fine-tuning with a small batch size 32. The next two rows correspond to fine-tuning with batch size 1024 using tuned learning rates vs. using the scaling rule. The last two rows represent fine-tuning using ScaLA with batch size 1024, also using tuned learning rates vs. the scaling rule. Even with square-root scaling, the large-batch baseline still cannot reach the small-batch accuracy (88.7 vs. 89.4). Moreover, although tuning the learning rates lead to better results on some datasets such as MNLI-m (84.9 vs. 85.1) and SST-2 (92.9 vs. 93.5), the square-root scaling rule leads to better results on other tasks such as QNLI (90.8 vs. 90) and QQP (91.4/88.4 vs. 90.9/87.7). So the best learning rates on fine-tuning tasks are not exactly sqrt. However, given that ScaLA with square-root learning rate scaling achieves on average better results than the grid search of learning rates (89.4 vs. 89.7), we suggest to use sqrt scaling for learning rates to simplify the hyperparameter tuning effort for ScaLA.

Table 6: Evaluation results on hyperparameter tuning vs. using square-root learning rate scaling.

|  | MNLI-m | QNLI | QQP | SST-2 | Avg |
|---|---|---|---|---|---|
| Bsz=32 (tuned, baseline) | 84.8 | 90.6 | 91/88 | 93.1 | 89.4 |
| Bsz=1024 (tuned, baseline) | 84.3 | 89.3 | 89.6/86.1 | 93 | 88.5 |
| Bsz=1024 (scaling rule, baseline) | 83.9 | 89.2 | 90.6/87.4 | 92.5 | 88.7 |
| Bsz=1024 (tuned, ScaLA) | 85.1 | 90 | 90.9/87.7 | 93.5 | 89.4 |
| Bsz=1024 (scaling rule, ScaLA) | 84.9 | 90.8 | 91.4/88.4 | 92.9 | 89.7 |

## D  CONVERGENCE ANALYSIS

In this section, we provide the formal statements and detailed proofs for the convergence rate. The convergence analysis builds on techniques and results in (Davis & Drusvyatskiy, 2018; You et al., 2019b). We consider the general problem of a two-player sequential game represented as nonconvex-nonconcave minimax optimization that is stochastic with respect to the outer (first) player playing $x \in \mathbb{X}$ while sampling $\xi$ from $Q$ and deterministic with respect to the inner (second) player playing $y \in \mathbb{Y}$, i.e.,

$$\min_x \max_y \mathbb{E}_{\xi \sim Q}[f(x, y, \xi)] := \min_x \mathbb{E}_{\xi \sim Q}[g(x, \xi)] \tag{2}$$

Since finding the Stackelberg equilibrium, i.e., the global solution to the saddle point problem, is NP-hard, we consider the optimality notion of a *local minimax* point (Jin et al., 2020). Since maximizing over $y$ may result in a non-smooth function even when $f$ is smooth, the norm of the gradient is not particularly a suitable metric to track the convergence progress of an iterative minimax optimization procedure. Hence, we use the gradient of the *Moreau envelope* (Davis & Drusvyatskiy, 2019) as the appropriate potential function. Let $\mu \in \mathbb{R}_+^h$. The $\mu$-Moreau envelope for a function $g : \mathbb{X} \to \mathbb{R}$ is defined as $g_\mu(x) := \min_z g(z) + \sum_{i=1}^h \frac{1}{2\mu^i}\|x^i - z^i\|^2$. Another reason for the choice of this potential function is due to the special property (Rockafellar, 2015) of the Moreau envelope that if its gradient $\nabla_x[g_\mu(x)]$ almost vanishes at $x$, such $x$ is close to a stationary point of the original function $g$.

**Assumptions:** We assume that $\mathbb{X} = \bigsqcup_{i=1}^h \mathbb{X}^i$ is partitioned into $h$ disjoint groups , i.e., in terms of training a neural network, we can think of the network having the parameters partitioned into $h$ (hidden) layers. The measure $Q$ characterizes the training data. Let $\widehat{\nabla}_x f(x, y)$ denote the noisy estimate of the true gradient $\nabla_x f(x, y)$. We assume that the noisy gradients are unbiased, i.e., $\mathbb{E}[\widehat{\nabla}_x f(x, y)] = \nabla_x f(x, y)$. For each group $i \in [h]$, we make the standard (groupwise) boundedness assumption (Ghadimi & Lan, 2013) on the variance of the stochastic gradients, i.e., $\mathbb{E}\|\widehat{\nabla}_x^i f(x, y) - \nabla_x^i f(x, y)\|^2 \le \sigma_i^2, \forall i \in [h]$. We assume that $f(x, y)$ has Lipschitz continuous gradients. Specifically, let $f(x, y)$ be $\alpha$-smooth in $x$ where $\alpha := (\alpha_1, \dots, \alpha_h)$ denotes the $h$-dimensional vector of (groupwise) Lipschitz parameters, i.e., $\|\nabla_x^i f(x_a, y) - \nabla_x^i f(x_b, y)\| \le \alpha_i \|x_a^i - x_b^i\|$, $\forall i \in [h]$ and $x_a, x_b \in \mathbb{X}, y \in \mathbb{Y}$. Let $\kappa_\alpha := \frac{\max_i \alpha_i}{\min_i \alpha_i}$.

Super-scripts are used to index into a vector ($i$ denotes the group index and $j$ denotes an element in group $i$). For any $c \in \mathbb{R}$, the function $\nu : \mathbb{R} \to [\mathcal{L}, \mathcal{U}]$ clips its values, i.e., $\nu(c) := \max(\mathcal{L}, \min(c, \mathcal{U}))$ where $\mathcal{L} < \mathcal{U}$. Let $\|.\|, \|.\|_1$ and $\|.\|_\infty$ denote the $\ell_2, \ell_1$, and $\ell_\infty$ norms. We assume that the true gradients are bounded, i.e., $\|\nabla_x f(x, y)\|_\infty \le \mathcal{G}$.

First, we begin with relevant supporting lemmas. The following lemma characterizes the convexity of an additive modification of $g$.

**Lemma 1** ((Lin et al., 2020; Jin et al., 2020; Rafique et al., 2021))**.** *Let* $g(x) := \max_y f(x, y)$ *with* $f$ *being* $\alpha$-*smooth in* $x$ *where* $\alpha \in \mathbb{R}^h_+$ *is the vector of groupwise Lipschitz parameters. Then,* $g(x) + \sum_{i=1}^h \frac{\alpha_i}{2} \|x^i\|^2$ *is convex in* $x$.

The following property of the Moreau envelope relates it to the original function.

**Lemma 2** ((Rockafellar, 2015))**.** *Let* $g$ *be defined as in Lemma 1. Let* $\widehat{x} = \arg\min_{\widetilde{x}} g(\widetilde{x}) + \sum_{i=1}^h \frac{1}{2\mu^i} \|\widetilde{x}^i - x^i\|^2$. *Then,* $\|g_\mu(x)\| \leq \epsilon$ *implies* $\|\widehat{x} - x\| \leq \|\mu\|_\infty \epsilon$ *and* $\min_h \|h\| \leq \epsilon$ *with* $h \in \partial g$ *where* $\partial g$ *denotes the subdifferential of* $g$.

We now present the formal version of Theorem 1 in Theorem 2. Note that Lemma 2 facilitates giving the convergence guarantees in terms of the gradient of the Moreau envelope. Recall that $t \in [T]$ denotes the epochs corresponding to the outer maximization. Without loss of generality, we set the delay parameter for injection of the adversarial perturbation in Algorithm 1 as $t_s = 0$. Here, we assume that the PGA provides an $\epsilon$-approximate maximizer.

**Theorem 2** (Groupwise outer minimization with an $\epsilon$-approximate inner maximization oracle)**.** *Let us define relevant constants as* $\mathcal{D} := \left(g_{1/2\alpha}(x_0) - \mathbb{E}(\min_x g(x))\right)$ *being the optimality gap due to initialization,* $\kappa_\alpha := \frac{\max_i \alpha_i}{\min_i \alpha_i}$ *being the condition number,* $\|\nabla_x f(x, y)\|_\infty \leq \mathcal{G}$ *being gradient bound,* $\mathcal{Z} := \max_{i,j,t} \frac{(\widehat{x}_t^{i,j} - x_t^{i,j})}{(\nabla_t^{i,j})} \sigma_i$ *being the variance term,* $\mathcal{L}, \mathcal{U}$ *being clipping constants such that* $\mathcal{L} \leq \mathcal{U}$. *For the outer optimization, setting the learning rate as* $\eta = \frac{1}{\mathcal{U}\sqrt{T}}$ *and scaling batch size as* $b = \frac{16T\mathcal{L}^2\mathcal{Z}^2}{\mathcal{U}^2}$, *we have*

$$\mathbb{E}\left[\|\nabla g_{1/2\alpha}(\overline{x})\|^2\right] \leq 4\epsilon\|\alpha\|_\infty + \frac{2\kappa_\alpha \mathcal{D}\mathcal{G}}{\sqrt{T}} \tag{3}$$

*where* $\overline{x}$ *is the estimator obtained from running* $T$ *steps of Algorithm 1 and picking* $x_t$ *uniformly at random for* $t \in [T]$.

*Proof.* In this proof, for brevity, we define the vector $\nabla_t := \nabla_x f(x, y)$, i.e., the gradient of the objective with respect to $x$, evaluated at the outer step $t$. Since evaluating gradients using mini-batches produces noisy gradients, we use $\widehat{\nabla}$ to denote the noisy version of a true gradient $\nabla$, i.e., $\widehat{\nabla} = \nabla + \Delta$ for a noise vector $\Delta$. For any outer step $t$, we have $f(x_t, \widehat{y}) \geq g(x_t) - \epsilon$ where $\widehat{y}$ is an $\epsilon$-approximate maximizer. For any $\widetilde{x} \in \mathbb{X}$, using the smoothness property (Lipschitz gradient) of $f$, we have

$$g(\widetilde{x}) \geq f(\widetilde{x}, y_t)$$

$$\geq f(x_t, y_t) + \sum_{i=1}^h \langle \nabla_t^i, \widetilde{x}^i - x_t^i \rangle - \sum_{i=1}^h \frac{\alpha_i}{2} \|\widetilde{x}^i - x_t^i\|^2$$

$$\geq g(x_t) - \epsilon + \sum_{i=1}^h \langle \nabla_t^i, \widetilde{x}^i - x_t^i \rangle - \sum_{i=1}^h \frac{\alpha_i}{2} \|\widetilde{x}^i - x_t^i\|^2 \tag{4}$$

Let $\phi_\mu(x, z) := g(z) + \sum_{i=1}^h \frac{1}{2\mu^i} \|x^i - z^i\|^2$. Recall that the $\mu$-Moreau envelope for $g$ is defined as $g_\mu(x) := \min_z \phi_\mu(x, z)$ and its gradient is the groupwise proximal operator given by $\nabla_x[g_\mu(x)] = \left[\frac{1}{\mu^1}\left(x^1 - \arg\min_{z^1} \phi_\mu(x, z)\right), \ldots, \frac{1}{\mu^h}\left(x^h - \arg\min_{z^h} \phi_\mu(x, z)\right)\right]$.

Now, let $\widehat{x}_t = \arg\min_x \phi_{1/2\alpha}(x_t, x) = \arg\min_x \left(g(x) + \sum_{i=1}^h \alpha_i \|x_t^i - x^i\|^2\right)$. Then, plugging in the update rule for $x$ at step $t + 1$ in terms of quantities at step $t$, using the shorthand $\nu_t^i := \nu(\|x_t^i\|)$

and conditioning on the filtration up to time $t$, we have

$$g_{1/2\alpha}(x_{t+1}) \leq g(\widehat{x}_t) + \sum_{i=1}^{h} \alpha_i \|x_{t+1}^i - \widehat{x}_t^i\|^2$$

$$\leq g(\widehat{x}_t) + \sum_{i=1}^{h} \alpha_i \left\| x_t^i - \eta_t \nu_t^i \frac{\widehat{\nabla}_t^i}{\|\widehat{\nabla}_t^i\|} - \widehat{x}_t^i \right\|^2$$

$$\leq g(\widehat{x}_t) + \sum_{i=1}^{h} \alpha_i \left\| x_t^i - \widehat{x}_t^i \right\|^2 + \sum_{i=1}^{h} 2\alpha_i \eta_t \left\langle \nu_t^i \frac{\widehat{\nabla}_t^i}{\|\widehat{\nabla}_t^i\|}, \widehat{x}_t^i - x_t^i \right\rangle + \sum_{i=1}^{h} \alpha_i \eta_t^2 (\nu_t^i)^2$$

$$\leq g_{1/2\alpha}(x_t) + \sum_{i=1}^{h} 2\alpha_i \eta_t \left\langle \nu_t^i \frac{\widehat{\nabla}_t^i}{\|\widehat{\nabla}_t^i\|}, \widehat{x}_t^i - x_t^i \right\rangle + \sum_{i=1}^{h} \alpha_i \eta_t^2 (\nu_t^i)^2$$

$$\leq g_{1/2\alpha}(x_t) + 2\eta_t \sum_{i=1}^{h} \alpha_i \nu_t^i \sum_{j=1}^{d_i} \left( \frac{\widehat{\nabla}_t^{i,j}}{\|\widehat{\nabla}_t^i\|} - \frac{\nabla_t^{i,j}}{\|\nabla_t^i\|} + \frac{\nabla_t^{i,j}}{\|\nabla_t^i\|} \right) \times (\widehat{x}_t^{i,j} - x_t^{i,j}) + \sum_{i=1}^{h} \alpha_i \eta_t^2 (\nu_t^i)^2$$

$$\leq g_{1/2\alpha}(x_t) + 2\eta_t \sum_{i=1}^{h} \alpha_i \nu_t^i \sum_{j=1}^{d_i} \left( \frac{\nabla_t^{i,j}}{\|\nabla_t^i\|} \right) \times (\widehat{x}_t^{i,j} - x_t^{i,j})$$

$$+ 2\eta_t \sum_{i=1}^{h} \alpha_i \nu_t^i \sum_{j=1}^{d_i} \left( \frac{\widehat{\nabla}_t^{i,j}}{\|\widehat{\nabla}_t^i\|} - \frac{\nabla_t^{i,j}}{\|\nabla_t^i\|} \right) \times (\widehat{x}_t^{i,j} - x_t^{i,j}) + \sum_{i=1}^{h} \alpha_i \eta_t^2 (\nu_t^i)^2$$

$$\leq g_{1/2\alpha}(x_t) + 2\eta_t \sum_{i=1}^{h} \frac{\alpha_i \nu_t^i}{\|\nabla_t^i\|} \left\langle \nabla_t^i, \widehat{x}_t^i - x_t^i \right\rangle$$

$$+ 2\eta_t \sum_{i=1}^{h} \alpha_i \nu_t^i \sum_{j=1}^{d_i} \left( \frac{\nabla_t^{i,j} + \Delta_t^{i,j}}{\|\nabla_t^i + \Delta_t^i\|} - \frac{\nabla_t^{i,j}}{\|\nabla_t^i\|} \right) \times (\widehat{x}_t^{i,j} - x_t^{i,j}) + \sum_{i=1}^{h} \alpha_i \eta_t^2 (\nu_t^i)^2$$

$$\leq g_{1/2\alpha}(x_t) + 2\eta_t \mathcal{U} \sum_{i=1}^{h} \frac{\alpha_i}{\|\nabla_t^i\|} \left\langle \nabla_t^i, \widehat{x}_t^i - x_t^i \right\rangle$$

$$+ 2\eta_t \sum_{i=1}^{h} \alpha_i \nu_t^i \sum_{j=1}^{d_i} \left( \frac{\|\nabla_t^i\|(\nabla_t^{i,j})(\nabla_t^{i,j} + \Delta_t^{i,j}) - \|\nabla_t^i + \Delta_t^i\|(\nabla_t^{i,j})^2}{\|\nabla_t^i + \Delta_t^i\|\|\nabla_t^i\|} \right) \times \frac{(\widehat{x}_t^{i,j} - x_t^{i,j})}{(\nabla_t^{i,j})}$$

$$+ \sum_{i=1}^{h} \alpha_i \eta_t^2 (\nu_t^i)^2$$

$$\overset{E_1}{\leq} g_{1/2\alpha}(x_t) + 2\eta_t \mathcal{U} \max_i \frac{\alpha_i}{\|\nabla_t^i\|} \left( g(\widehat{x}_t) - g(x_t) + \epsilon + \sum_{i=1}^{h} \frac{\alpha_i}{2} \|\widehat{x}^i - x_t^i\|^2 \right)$$

$$+ 2\eta_t \sum_{i=1}^{h} \alpha_i \nu_t^i \max_j \frac{(\widehat{x}_t^{i,j} - x_t^{i,j})}{(\nabla_t^{i,j})} \left( \frac{\langle \nabla_t^i, \nabla_t^i + \Delta_t^i \rangle - \|\nabla_t^i + \Delta_t^i\|\|\nabla_t^i\|}{\|\nabla_t^i + \Delta_t^i\|} \right) + \sum_{i=1}^{h} \alpha_i \eta_t^2 (\nu_t^i)^2$$

$$\leq g_{1/2\alpha}(x_t) + 2\eta_t \mathcal{U} \max_i \frac{\alpha_i}{\|\nabla_t^i\|} \left( g(\widehat{x}_t) - g(x_t) + \epsilon + \sum_{i=1}^{h} \frac{\alpha_i}{2} \|\widehat{x}^i - x_t^i\|^2 \right)$$

$$- 2\eta_t \sum_{i=1}^{h} \alpha_i \nu_t^i \max_j \frac{(\widehat{x}_t^{i,j} - x_t^{i,j})}{(\nabla_t^{i,j})} \left( \frac{\|\nabla_t^i + \Delta_t^i\|\|\nabla_t^i\| - \|\nabla_t^i + \Delta_t^i\|^2 + \langle \Delta_t^i, \nabla_t^i + \Delta_t^i \rangle}{\|\nabla_t^i + \Delta_t^i\|} \right)$$

$$+ \sum_{i=1}^{h} \alpha_i \eta_t^2 (\nu_t^i)^2 \tag{5}$$

$$\leq g_{1/2\alpha}(x_t) + 2\eta_t \mathcal{U} \max_i \frac{\alpha_i}{\|\nabla_t^i\|} \left( g(\widehat{x}_t) - g(x_t) + \epsilon + \sum_{i=1}^{h} \frac{\alpha_i}{2} \|\widehat{x}^i - x_t^i\|^2 \right)$$

$$- 2\eta_t \sum_{i=1}^{h} \alpha_i \nu_t^i \max_j \frac{(\widehat{x}_t^{i,j} - x_t^{i,j})}{(\nabla_t^{i,j})} \left( \|\nabla_t^i\| - \|\nabla_t^i + \Delta_t^i\| - \frac{|\langle \Delta_t^i, \nabla_t^i + \Delta_t^i \rangle|}{\|\nabla_t^i + \Delta_t^i\|} \right) + \sum_{i=1}^{h} \alpha_i \eta_t^2 (\nu_t^i)^2$$

$$\overset{E_2}{\leq} g_{1/2\alpha}(x_t) + 2\eta_t \mathcal{U} \max_i \frac{\alpha_i}{\|\nabla_t^i\|} \left( g(\widehat{x}_t) - g(x_t) + \epsilon + \sum_{i=1}^{h} \frac{\alpha_i}{2} \|\widehat{x}^i - x_t^i\|^2 \right)$$

$$- 2\eta_t \sum_{i=1}^{h} \alpha_i \nu_t^i \max_j \frac{(\widehat{x}_t^{i,j} - x_t^{i,j})}{(\nabla_t^{i,j})} \left( \|\nabla_t^i\| - \|\nabla_t^i + \Delta_t^i\| - \|\Delta_t^i\| \right) + \sum_{i=1}^{h} \alpha_i \eta_t^2 (\nu_t^i)^2$$

$$\overset{E_3}{\leq} g_{1/2\alpha}(x_t) + 2\eta_t \mathcal{U} \max_i \frac{\alpha_i}{\|\nabla_t^i\|} \left( g(\widehat{x}_t) - g(x_t) + \epsilon + \sum_{i=1}^{h} \frac{\alpha_i}{2} \|\widehat{x}^i - x_t^i\|^2 \right)$$

$$- 4\eta_t \sum_{i=1}^{h} \alpha_i \nu_t^i \max_j \frac{(\widehat{x}_t^{i,j} - x_t^{i,j})}{(\nabla_t^{i,j})} \|\Delta_t^i\| + \sum_{i=1}^{h} \alpha_i \eta_t^2 (\nu_t^i)^2$$

$$g_{1/2\alpha}(x_T) \overset{E_4}{\leq} g_{1/2\alpha}(x_0) + 2\mathcal{U} \sum_{t=0}^{T-1} \eta_t \max_i \frac{\alpha_i}{\|\nabla_t^i\|} \left( g(\widehat{x}_t) - g(x_t) + \epsilon + \sum_{i=1}^{h} \frac{\alpha_i}{2} \|\widehat{x}^i - x_t^i\|^2 \right)$$

$$- 4 \sum_{t=0}^{T-1} \eta_t \sum_{i=1}^{h} \alpha_i \nu_t^i \max_j \frac{(\widehat{x}_t^{i,j} - x_t^{i,j})}{(\nabla_t^{i,j})} \|\Delta_t^i\| + \sum_{t=0}^{T-1} \sum_{i=1}^{h} \alpha_i \eta_t^2 (\nu_t^i)^2$$

where we have used Hölder's inequality along with bound equation 4 in $E_1$, Cauchy-Schwarz inequality in $E_2$, triangle inequality in $E_3$, telescoping sum in $E_4$. Rearranging and using $\eta_t = \eta$ in

$E_5$ along with Hölder's inequality,

$$\frac{1}{2\eta\mathcal{U}}\left(g_{1/2\alpha}(x_T) - g_{1/2\alpha}(x_0)\right) \le \sum_{t=0}^{T-1} \max_i \frac{\alpha_i}{\|\nabla_t^i\|}\left(g(\widehat{x}_t) - g(x_t) + \epsilon + \sum_{i=1}^{h}\frac{\alpha_i}{2}\|\widehat{x}^i - x_t^i\|^2\right)$$

$$-\frac{2}{\mathcal{U}}\sum_{t=0}^{T-1}\sum_{i=1}^{h}\alpha_i\nu_t^i \max_j \frac{(\widehat{x}_t^{i,j} - x_t^{i,j})}{(\nabla_t^{i,j})}\|\Delta_t^i\| + \frac{\eta}{2\mathcal{U}}\sum_{t=0}^{T-1}\sum_{i=1}^{h}\alpha_i(\nu_t^i)^2$$

$$\frac{1}{2\eta\mathcal{U}}\left(g_{1/2\alpha}(x_T) - g_{1/2\alpha}(x_0)\right) \overset{E_5}{\le} \max_{i,t}\frac{\alpha_i}{\|\nabla_t^i\|}\sum_{t=0}^{T-1}\left(g(\widehat{x}_t) - g(x_t) + \epsilon + \sum_{i=1}^{h}\frac{\alpha_i}{2}\|\widehat{x}^i - x_t^i\|^2\right)$$

$$-\frac{2}{\mathcal{U}}\sum_{t=0}^{T-1}\sum_{i=1}^{h}\alpha_i\nu_t^i \max_j \frac{(\widehat{x}_t^{i,j} - x_t^{i,j})}{(\nabla_t^{i,j})}\|\Delta_t^i\| + \frac{\eta}{2\mathcal{U}}\sum_{t=0}^{T-1}\sum_{i=1}^{h}\alpha_i(\nu_t^i)^2$$

Dividing by $T$ and rearranging,

$$\frac{1}{T}\sum_{t=0}^{T-1}\left(g(x_t) - g(\widehat{x}_t) - \sum_{i=1}^{h}\frac{\alpha_i}{2}\|\widehat{x}^i - x_t^i\|^2\right) \le \epsilon - \frac{1}{2\eta\mathcal{U}\zeta T}\left(g_{1/2\alpha}(x_T) - g_{1/2\alpha}(x_0)\right)$$

$$-\frac{2}{\mathcal{U}\zeta T}\sum_{t=0}^{T-1}\sum_{i=1}^{h}\alpha_i\nu_t^i \max_j \frac{(\widehat{x}_t^{i,j} - x_t^{i,j})}{(\nabla_t^{i,j})}\|\Delta_t^i\|$$

$$+\frac{\eta}{2\mathcal{U}\zeta T}\sum_{i=1}^{h}\alpha_i\sum_{t=0}^{T-1}(\nu_t^i)^2$$

where we define $\zeta := \max_{i,t}\frac{\alpha_i}{\|\nabla_t^i\|}$. Defining $\mathcal{D} := \left(g_{1/2\alpha}(x_0) - \mathbb{E}(\min_x g(x))\right)$ and taking expectation with respect to $\xi$ on both sides, we have

$$\frac{1}{T}\sum_{t=0}^{T-1}\mathbb{E}\left(g(x_t) - g(\widehat{x}_t) - \sum_{i=1}^{h}\frac{\alpha_i}{2}\|\widehat{x}^i - x_t^i\|^2\right) \le \epsilon + \frac{\mathcal{D}}{2\eta\mathcal{U}\zeta T}$$

$$-\frac{2\mathcal{L}}{\mathcal{U}\zeta T}\sum_{t=0}^{T-1}\sum_{i=1}^{h}\alpha_i \max_j \frac{(\widehat{x}_t^{i,j} - x_t^{i,j})}{(\nabla_t^{i,j})}\mathbb{E}\|\Delta_t^i\| + \frac{\eta\mathcal{U}\|\alpha\|_1}{2\zeta}$$

$$\overset{E_6}{\le} \epsilon + \frac{\mathcal{D}}{2\eta\mathcal{U}\zeta T}$$

$$-\frac{2\mathcal{L}}{\mathcal{U}\zeta T}\sum_{t=0}^{T-1}\sum_{i=1}^{h}\alpha_i \max_j \frac{(\widehat{x}_t^{i,j} - x_t^{i,j})}{(\nabla_t^{i,j})}\frac{\sigma_i}{\sqrt{b}} + \frac{\eta\mathcal{U}\|\alpha\|_1}{2\zeta}$$

$$\overset{E_7}{\le} \epsilon + \frac{\mathcal{D}}{2\eta\mathcal{U}\zeta T}$$

$$-\frac{2\mathcal{L}\|\alpha\|_1}{\mathcal{U}\zeta\sqrt{b}}\max_{i,j,t}\frac{(\widehat{x}_t^{i,j} - x_t^{i,j})}{(\nabla_t^{i,j})}\sigma_i + \frac{\eta\mathcal{U}\|\alpha\|_1}{2\zeta}$$

$$\overset{E_8}{=} \epsilon + \frac{\mathcal{D}}{2\eta\mathcal{U}\zeta T} - \frac{2\mathcal{L}\|\alpha\|_1\mathcal{Z}}{\mathcal{U}\zeta\sqrt{b}} + \frac{\eta\mathcal{U}\|\alpha\|_1}{2\zeta} \qquad (6)$$

where we have used the assumption on the variance of stochastic gradients in $E_6$, Hölder's inequality in $E_7$ and we define $\mathcal{Z} := \max_{i,j,t}\frac{(\widehat{x}_t^{i,j} - x_t^{i,j})}{(\nabla_t^{i,j})}\sigma_i$ in $E_8$; $b$ denotes batch size. Now, we lower bound

the left hand side using the convexity of the additive modification of $g$.

$$g(x_t) - g(\widehat{x}_t) - \sum_{i=1}^{h} \frac{\alpha_i}{2} \|\widehat{x}^i - x_t^i\|^2$$

$$\geq g(x_t) + 0 - g(\widehat{x}_t) - \sum_{i=1}^{h} \alpha_i \|\widehat{x}^i - x_t^i\|^2 + \sum_{i=1}^{h} \frac{\alpha_i}{2} \|\widehat{x}^i - x_t^i\|^2$$

$$\geq \left( \left( g(x_t) + \sum_{i=1}^{h} \alpha_i \|x_t^i - x_t^i\|^2 \right) - \min_x \left( g(x_t) + \sum_{i=1}^{h} \alpha_i \|x^i - x_t^i\|^2 \right) \right) + \sum_{i=1}^{h} \frac{\alpha_i}{2} \|\widehat{x}^i - x_t^i\|^2$$

$$\geq \sum_{i=1}^{h} \frac{\alpha_i}{2} \|\widehat{x}^i - x_t^i\|^2 + \sum_{i=1}^{h} \frac{\alpha_i}{2} \|\widehat{x}^i - x_t^i\|^2 = \sum_{i=1}^{h} \frac{4\alpha_i^2}{4\alpha_i} \|\widehat{x}^i - x_t^i\|^2$$

$$\overset{E_9}{\geq} \frac{1}{4 \max_i \alpha_i} \|\nabla g_{1/2\alpha}(x_t)\|^2 \tag{7}$$

where we have used the expression for the gradient of the Moreau envelope in $E_9$. Combining the inequalities from Equation equation 7 and Equation equation 6, we have

$$\frac{1}{T} \sum_{t=0}^{T-1} \mathbb{E} \left( \frac{1}{4 \max_i \alpha_i} \|\nabla g_{1/2\alpha}(x_t)\|^2 \right) \leq \epsilon + \frac{\mathcal{D}}{2\eta \mathcal{U} \zeta T} + \left( \frac{\eta \mathcal{U}}{2\zeta} - \frac{2\mathcal{L}\mathcal{Z}}{\mathcal{U}\zeta\sqrt{b}} \right) \|\alpha\|_1$$

Setting the learning rate as $\eta = \frac{1}{\mathcal{U}\sqrt{T}}$ and batch size as $b = \frac{16T\mathcal{L}^2\mathcal{Z}^2}{\mathcal{U}^2}$,

$$\frac{1}{T} \sum_{t=0}^{T-1} \mathbb{E} \left[ \|\nabla g_{1/2\alpha}(x_t)\|^2 \right] \leq 4\epsilon \max_i \alpha_i + \frac{2\mathcal{D} \max_i \alpha_i}{\zeta\sqrt{T}}$$

Now, to simplify $\zeta$, using the inequality that $\max_k(a_k \cdot b_k) \geq \min_{k_a} a_{k_a} \cdot \min_{k_b} b_{k_b}$ for two finite sequences $\{a, b\}$ with positive values, along with the bounded gradients assumption, we have

$$\frac{1}{T} \sum_{t=0}^{T-1} \mathbb{E} \left[ \|\nabla g_{1/2\alpha}(x_t)\|^2 \right] \leq 4\epsilon \max_i \alpha_i + \frac{2\mathcal{D}\mathcal{G} \max_i \alpha_i}{\sqrt{T} \min_i \alpha_i} = 4\epsilon \|\alpha\|_\infty + \frac{2\kappa_\alpha \mathcal{D}\mathcal{G}}{\sqrt{T}}$$

where $\kappa_\alpha := \frac{\max_i \alpha_i}{\min_i \alpha_i}$. $\qquad \square$

In analyzing inexact version, as in Theorem 2, we assumed the availability of an adversarial oracle. Next, we open up the adversarial oracle to characterize the oracle-free complexity. In order to do this, we will assume additional properties about the function $f$ as well as our deterministic perturbation space, $\mathbb{Y}_t \subseteq \mathbb{Y}, \forall t \in [T]$. Note that, for any given $t$, $y_\tau \in \mathbb{Y}_t, \forall \tau \in \mathcal{T}$. We recall the following guarantee for generalized non-convex projected gradient ascent.

**Lemma 3** ((Jain & Kar, 2017))**.** *For every $t$, Let $f(x_t, \cdot)$ satisfy restricted strong convexity with parameter $\mathcal{C}$ and restricted strong smoothness with parameter $\mathcal{S}$ over a non-convex constraint set with $\mathcal{S}/\mathcal{C} < 2$, ie, $\frac{\mathcal{C}}{2}\|z - y\|^2 \leq f(x_t, y) - f(x_t, z) - \langle \nabla_z f(x_t, z), y - z \rangle \leq \frac{\mathcal{S}}{2}\|z - y\|^2$ for $y, z \in \mathbb{Y}_t$. For any given $t$, let the PGA-$\mathcal{T}$ algorithm $y_\tau \leftarrow \Pi_\epsilon[y_{\tau-1} + \rho \nabla_y f(x_t, y)]$ be executed with step size $\rho = 1/\mathcal{S}$. Then after at most $\mathcal{T} = O\left( \frac{\mathcal{C}}{2\mathcal{C}-\mathcal{S}} \log \frac{1}{\epsilon} \right)$ steps, $f(x_t, y_\mathcal{T}) \geq \max_y f(x_t, y) - \epsilon$.*

Using Theorem 2 and Lemma 3 (together with the additional restricted strong convexity/smoothness assumptions), we have the following theorem on the full oracle-free rates for Algorithm 1.

**Theorem 3** (Groupwise outer minimization with inner maximization using projected gradient ascent)**.** *Setting the inner iteration count as $\mathcal{T} = O\left( \frac{\mathcal{C}}{2\mathcal{C}-\mathcal{S}} \log \frac{8\|\alpha\|_\infty}{\epsilon} \right)$ and the outer iteration count as $T = \frac{16\kappa_\alpha \mathcal{D}^2 \mathcal{G}^2}{\epsilon^2}$, for a combined total of $O(\frac{1}{\epsilon^2} \log \frac{1}{\epsilon})$ adaptive adversarial iterations, Algorithm 1 achieves $\mathbb{E}\left[ \|\nabla g_{1/2\alpha}(\overline{x})\|^2 \right] \leq \epsilon$.*

