# OpenReview forum: "ScaLA: Speeding-Up Fine-tuning of Pre-trained Transformer Networks via Efficient and Scalable Adversarial Perturbation"
_ICLR.cc/2022/Conference — ICLR 2022 Submitted_

### Official Review · Reviewer_FdJg · 2021-10-27

**Correctness:** 4
**Technical Novelty And Significance:** 2
**Empirical Novelty And Significance:** 3
**Recommendation:** 6
**Confidence:** 3

**Main Review:**

**Strong Points:**
- This paper is well organized and it's good to see that we can also use large batch training for the fine-tuning process.
- The authors did plenty of ablation studies to break down the impact of each component in ScaLA, and it's interesting that using delayed perturbation just has minor side effects.

**Weak Points and Questions:**
- From Table 1, ScaLA even matches or surpass the performance of small batch training, I'm curious about the performance of similar adversarial training algorithm when applied on small batch fine-tuning.
- In Algorithm 1, should line 9 also be different for classification and regression task?
- It would be better if the author can show the results of baselines when using the same fine-tuning time. Table 1 shows that ScaLA performs similar to small batch baselines but is faster, Table 2 shows that ScaLA achieves higher accuracy but is slower than large batch baselines. I would like to know the result of baseline under the same training cost (e.g., using a batch size of 512).

**Summary Of The Paper:**

This paper proposes an adversarial training algorithm to speed up the fine-tuning of large Transformer models while retaining strong performance. The authors also incorporate several tricks such as delayed perturbation to lower the computational overhead.

**Summary Of The Review:**

In summary, this paper is well-written and i can see the efforts of the authors to make it rigorous and well-organized, the ablation studies and analysis are also very comprehensive. But I'm not sure about the novelty of this paper, ScaLA to me is more like a combination of several methods, incorporating adversarial perturbation and layer-wise optimizer. Moreover, the benefits of adversarial training on normal batch size in both NLP and CV domains are well-studied, so I think the novelty is restricted. I vote for a weak accept for the written quality and the rigorous analysis of this paper.

---

> ### Author Response · Authors · 2021-11-21
> **Response to Reviewer FdJg**
>
> Thank you for taking the time to review our submission.
>
> Small batch adversarial training: For small batch fine-tuning with adversarial training, please refer to [1] for detailed evaluation results.
>
> Algorithm 1 line 9: Yes, line 9 is for classification. For regression, it should be handled as noted in "practical consideration" in Section 3. We will add a footnote in the paper to clarify this distinction.
>
> Result of baseline under the same training cost: Figure 6 can help answer this question. Figure 6 shows the comparison of time-convergence results between ScaLA and the baseline on a batch size of 2K. If we compare the baseline results at epoch 12 and the ScaLA results at epoch 6, we can see that ScaLA still achieves better accuracy (85.1) than the baseline (84) despite having a small amount of training cost. This result shows that training the downstream tasks longer with the baseline method does not lead to better generalizability.
>
> [1] SMART: Robust and Efficient Fine-Tuning for Pre-trained Natural Language Models through Principled Regularized Optimization,  https://arxiv.org/pdf/1911.03437.pdf

---

### Official Review · Reviewer_jy9c · 2021-11-02

**Correctness:** 4
**Technical Novelty And Significance:** 3
**Empirical Novelty And Significance:** 3
**Recommendation:** 6
**Confidence:** 4

**Main Review:**

Pros:
1. Firstly, the paper is generally well written and easy to follow.
2. The proposed method contains several techniques to reduce the computational overhead from adversarial training, leading to significant training time savings.
2. The experimental results seem satisfactory. The convergence is faster compared to the baseline.

Cons:
1. Firstly, the core contribution, using adversarial training to improve generalization and convergence under large-batch training, has been adopted in NLP (FreeLb), which limits the novelty of the proposed paper. (A minor point, the comparison in Table 5 does not seem fair; ScaLA uses LARS optimizer for large-batch training, while FreeLb does not. This optimizer is not part of the technical contribution, and should be kept the same).
2. FreeLb can also be used for training from scratch, which is another important setting. It is unclear whether ScaLA can also be applied to this setting.
3. The authors argued that ScaLA can help to escape sharp minimum. Other techniques like the SAM optimizer [a] also have the same effect and proved to be working for ViTs [b]. It would be great to compare against these methods.
4. It is a bit unclear to me what are the training settings for each method. It seems that due to the different convergence speeds, different methods use different numbers of training epochs/iterations. How do you choose when to halt training to report the results (e.g., in Table 1)?


[a] Foret et al., Sharpness-aware minimization for efficiently improving generalization
[b] Chen et al., When Vision Transformers Outperform ResNets without Pre-training or Strong Data Augmentations


**Summary Of The Paper:**

In this paper, the authors propose ScaLA to speed up the fine-tuning of large pre-trained transformer language models. Specifically, ScaLA employs adversarial training to solve the worse convergence in large-batch training. Several techniques are proposed to reduce the computational overhead. Experiments on the GLUE benchmark show faster convergence compared to existing methods.

**Summary Of The Review:**

Generally, I feel this paper presents an interesting solution to solve the convergence and generalization problem of large-batch fine-tuning. However, the proposed method is somewhat similar to existing work, which limits the novelty. I would like to hear back from the authors for the final opinion.

---

> ### Author Response · Authors · 2021-11-21
> **Response to Reviewer jy9c**
>
> Thank you for taking the time to review our submission.
>
> Novelty to FreeLb: We politely argue that the novelty of our work to FreeLb lies in how ScaLA supports large-batch adversarial training much faster than FreeLb. As shown in Table 5, ScaLA is 2.6--2.8x faster than FreeLb to achieve a similar or slightly better accuracy. Furthermore, while using layer-wise adaptive learning rate (LARS) for large-batch adversarial training may feel obvious in hindsight, almost no research has been published in combining the two and evaluates to what extend layer-wise adaptive learning rate (e.g., LARS) may benefit large-batch adversarial training. Furthermore, no prior work studies the convergence of min-max optimization under adaptive learning rate, so an important aspect our analysis (Theorem 1 and Appendix D) brings out is the specific dependence on the batch size in stochastic-deterministic min-max optimization, which clearly differentiates our work from FreeLb.
>
> Training from scratch: The reviewer is correct that FreeLb can also be used for training from scratch (e.g., pre-training the masked language model pre-training from random initialization). However, to the best of our knowledge, FreeLb is only evaluated on fine-tuning adaption and has not been evaluated on pre-training from scratch either, presumably because pre-training for language models requires huge computing resources. As an example, while it takes 12 hours to fine-tune RoBERTa-large on MNLI, it takes approximately 32 GPU days to finish one pre-training experiment. Given this practical constraint, we cannot provide the comprehensive analysis as those reported in Table 1-3. Note that we also did a thorough hyperparameter sweep to ensure that the superior accuracy is an artifact of the technique rather than using sub-optimal hyperparameters, which could also be challenging to verify for pre-training.
>
> Comparison with SAM: We thank the reviewer for sharing the SAM optimizer, and we will cite it. We believe the novelty in work again
> derives from how our approach induces sharpness-aware optimization much more cheaply than SAM. SAM improves the generalization of large-batch training through sharpness-aware optimization. However, less consideration was given to computational requirements, which becomes a concern with more complex networks and bigger datasets. For example, SAM still requires adversarial training to be performed in the initial phase of training. A second novel contribution is our investigation into to what extend the fine-tuning of pre-trained language models can benefit from adversarial training from a training speed and cost perspective. Since large-scale Transformer-based language models are very different from image classification in computer vision, any technique that works on the latter is not guaranteed to work on the former. Thus we politely argue that works without evaluations on Transformer-based language models cannot be regarded as directly-comparable related works, but we definitely agree that it would be interesting to explore how to apply those methods to Transformer-based language models.
>
> When to halt: All methods stop after training the same number of samples (i.e., epochs). The difference in training iterations is caused by using different batch sizes per iteration (column 3).  We follow the best practice for choosing the batch sizes for the baselines, and we show that our approach allows us to scale up the batch sizes effectively such that it takes fewer iterations to process the same number of samples to achieve the same accuracy.

---

> > ### Comment · Reviewer_jy9c · 2021-11-25
> > **Response**
> >
> > Thanks for the reply.
> >
> > 1. Thanks for pointing out that FreeLb is also tested under the fine-tuning setting. It makes sense to study fine-tuning for now due to the limited computation budget.
> > Regarding the LARS optimizer, I think it would be still reasonable to clearly show the contribution of each component through the ablation study. That being said, comparing with FreeLb+LARS would be a necessary baseline to show the advantage of ScaLA itself over FreeLb.
> >
> >  2. SAM optimizer has been extensively validated on vision transformer models in [a]. Given that the ViT models have roughly the same architecture compared to the original transformer model, it is reasonable to believe that it will also work quite well. In [a], SAM can also get better accuracy at the same epochs (i.e., faster convergence). Given the similar methodology (incorporating adversarial training into optimization), I think a comparison would be helpful.
> >
> >
> > [a] Chen et al., When Vision Transformers Outperform ResNets without Pre-training or Strong Data Augmentations

---

> > > ### Author Response · Authors · 2021-11-25
> > > **Addressing comparison with related work**
> > >
> > > Thank you for taking the time to respond.
> > >
> > > Ablation study of LARS: As you note, it would be important to identify the contribution of each component through ablation studies. We did perform the ablation studies of LARS. In Section 4.2 Table 3,  we reported the results of our approach with and without LARS (w/o Groupwise LR). Across the tasks, we observe a small 0.2 points of accuracy drop on average when disabling LARS. We believe the ScaLA w/o LARS results provide a more direct comparison to the original FreeLb method, and our ablation studies show to what extent ScaLA benefits from LARS. Please let us know if this ablation study helps address your concern.
> > >
> > > Comparison with SAM:  We totally agree that it would be interesting to explore different optimizers such as SAM for Transformers on both vision and NLP tasks, which have not been extensively explored before. The commit history shows that the first version of [a] was posted on Jun 3rd and was recently updated on October 11th. That was after this manuscript was submitted to ICLR (October 5th). We thank the reviewer for sharing this work, and we will cite it. In a later version of this paper (not enough time to catch rebuttal deadline) we would also add this experiment if time permits. However, given that SAM has only been very recently applied to Transformers (e.g., ViT) on a different domain (e.g., computer vision) [a], we kindly ask the reviewer to please take into consideration of its contemporaneousness when making a judgment and not discouraging work that has been developed concurrently.

---

> > > > ### Comment · Reviewer_jy9c · 2021-11-27
> > > > **Response**
> > > >
> > > > Thanks for pointing out the existing comparison with FreeLb. According to the experimental results, the proposed method does have the merit of faster training. I am willing to raise my score to 6.

---

> > > > > ### Author Response · Authors · 2021-11-27
> > > > > **Thank you for the response**
> > > > >
> > > > > Dear reviewer,
> > > > >
> > > > > We are glad that we have addressed the raised concerns, and we thank you for devoting your time and effort to engage with us for this continual discussion.
> > > > >
> > > > > Best regards,
> > > > > Our team

---

### Official Review · Reviewer_io2b · 2021-11-08

**Correctness:** 3
**Technical Novelty And Significance:** 2
**Empirical Novelty And Significance:** 2
**Recommendation:** 5
**Confidence:** 3

**Details Of Ethics Concerns:**

No ethics concerns

**Main Review:**

The main idea of this paper is to introduce adversarial perturbations to improve the convergence rate of large batch fine-tuning of sequence models. The proposed idea is very simple and effective. In this paper, the authors demonstrate a nearly three times speedup in BERT and RoBERTa fine-tuning on the GLUE dataset. Although the observations made in the paper such as only one PGD step is sufficient for generating perturbation, the novelty of the idea is limited. As mentioned in the paper, both aspects of the idea have been proposed and validated for training convolutional neural networks. Therefore, a throughout evaluation of sequence models at different scale on several NLP datasets is critical. My major concerns are listed blow.

major:
1) Empirical results on other datasets: Since the goal of the paper is to speed up finetuning, it is important to validate whether the proposed method can be generalized to different datasets such as SQuAD v2.

2) Empirical results on large models: The paper is motivated by the increasing cost of finetuning large sequence models. However, the paper only includes results on BERTBase model. It is important to include more results on large sequence models such as BERTLarge, Google T5 Large [1], and GPT-2/3 models.

3) Concurrent adversarial learning: A recent work in [2] on improving the convergence rate of very large small batch training also suggests adding adversarial perturbations. More interestingly, they retain a delayed model to generate adversarial perturbations, which makes the process of training and generating adversarial perturbations completely parallel.

minor:
1) caption of Figure 2: with and with

[1] Exploring the Limits of Transfer Learning with a Unified Text-to-Text Transformer
[2] Concurrent Adversarial Learning for Large-Batch Training

**Summary Of The Paper:**

The authors propose introducing adversarial perturbation in the late phase of finetuning of sequence models to enable large mini-batch. The proposed method can speed up the finetuning by almost three times.

**Summary Of The Review:**

The authors proposed to add adversarial perturbation to enable large mini-batch finetuning of  sequence models. However, the empirical study is limited to BERTBase model on GLUE datasets.

---

> ### Author Response · Authors · 2021-11-21
> **Response to Reviewer io2b**
>
> Thank you for taking the time to review our submission.
>
> The paper includes results on BERT-base only: Actually, we did evaluate and report results on other models. For example, both Table 1 and Table 2 report results of RoBERTa-large, which is not only a larger auto-encoder model with 350M parameter but also more robustly optimized than BERT-large with more steps and larger corpora. Fine-tuning RoBERTa-large is much slower than fine-tuning BERT-based due to the increased model size, and ScaLA provides up to 12.8x faster fine-tuning speed to RoBERTa-large. These results show that our approach works beyond BERT-base.
>
> Results on other datasets: We agree with the reviewer's point that evaluating more tasks is useful! We initially focused on GLUE because it contains a variety of tasks and has been used as the main benchmark for evaluating pre-trained models. That said, we have since run experiments on SQuAD2.0 and will add the more comprehensive evaluation results to the final version of the paper. Results are consistent with the GLUE ones (e.g., ScaLA speeds up fine-tuning SQuDA by allowing fine-tuning with a much larger effective batch size per iteration while achieving comparable accuracy).
>
> Comparison with ConAdv: We thank the reviewer for sharing this work, and we will cite it. Although ConAdv and our work share a similar concept of using adversarial learning to increase the batch size in large-batch training, the techniques are quite different. ConAdv tradeoffs more hardware for parallel execution of the min-max optimization in adversarial training by decoupling the sequential execution of min-max with stale parameters. In contrast, we reduce the training cost of adversarial training through delayed projected gradient ascent and inexact inner maximization. Moreover, while ConAdv studies adversarial training for small models like ResNet and EfficientNet on image classification, our technique is designed and validated specifically for the problem of speeding up the adaption of large-scale pre-trained Transformer models. Finally, the ICLR reviewer instructions ( https://iclr.cc/Conferences/2022/ReviewerGuide) mention that "authors are encouraged to cite and discuss all relevant papers, but they may be excused for not knowing about papers not published in peer-reviewed conference proceedings or journals, which includes papers exclusively available on arXiv. "For this reason, the reviewer may want to reconsider using ConAdv as a weakness against our work and not discourage the publication of original ideas that have been developed independently and concurrently.

---

### Decision · Program_Chairs · 2022-01-20

**Decision:**

Reject

**Comment:**

The submission considers a method involving adversarial training to speed up the fine-tuning of large pre-trained transformer language models. Reviewers consider it to be a borderline paper.  Many suggestions are made by the reviewers which will help improve the presentation and substance and make it more useful for the community.